# Learning Models as Functionals of Signed-Distance Fields for Manipulation Planning

**Danny Driess**[1]    **Jung-Su Ha**[1]    **Marc Toussaint**[1]    **Russ Tedrake**[2]

[1]TU Berlin, Germany
[2]Massachusetts Institute of Technology, USA

**Abstract:** This work proposes an optimization-based manipulation planning framework where the objectives are learned functionals of signed-distance fields that represent objects in the scene. Most manipulation planning approaches rely on analytical models and carefully chosen abstractions/state-spaces to be effective. A central question is how models can be obtained from data that are not primarily accurate in their predictions, but, more importantly, enable efficient reasoning within a planning framework, while at the same time being closely coupled to perception spaces. We show that representing objects as signed-distance fields not only enables to learn and represent a variety of models with higher accuracy compared to point-cloud and occupancy measure representations, but also that SDF-based models are suitable for optimization-based planning. To demonstrate the versatility of our approach, we learn both kinematic and dynamic models to solve tasks that involve hanging mugs on hooks and pushing objects on a table. We can unify these quite different tasks within one framework, since SDFs are the common object representation. Video: https://youtu.be/ga8Wlkss7co

**Keywords:** Manipulation Planning, Signed Distance Fields, Model Learning

## 1   Introduction

Manipulation planning is challenging for multiple reasons. On the one hand, planning robot motions to solve a task can be formulated as a decision problem over a high-dimensional, non-convex space, including discrete and continuous aspects. Especially long-horizon tasks that consist of multiple manipulation steps have the property that motions have to be coordinated globally with the future goal. This coupling of potentially all variables requires joint reasoning and makes the problem particularly challenging [1]. On the other hand, the problem solving capabilities of a planning framework is inherently dependent on its underlying models. The field of Task and Motion Planning (TAMP) has made significant progress in solving challenging multi-step, long-horizon tasks [2], ranging from ones that involve mainly kinematic models [3, 4, 5, 6, 7] to dynamic tasks that require reasoning about forces, friction etc. based on more general dynamic equations [8, 9, 10, 11, 12, 13, 14]. However, most TAMP approaches rely on carefully chosen abstractions and analytically defined models in order to be successful and efficient. In particular, TAMP often makes simplifying assumptions on the possible geometries of objects it can deal with to define manipulation constraints in the first place. It is unclear how these models can be grounded from sensor information.

To overcome these issues, a natural idea is to replace the analytic models in TAMP frameworks with learned ones. Recent advances in deep learning have enabled to learn predictive forward models even in high-dimensional observation spaces like images. The typical objective for learning a forward model is its predictive accuracy. However, having an accurate model does not necessarily imply that a planning framework can utilize it efficiently. While having an accurate forward prediction model might be sufficient for short-horizon tasks, especially for long-horizon tasks, learned models can exhibit too high combinatorics for sampling or non-informative gradients for achieving future goals.

This paper aims to address these challenges by learning models that can be used effectively by a planning framework while at the same time using a general object representation more closely related to senors spaces. To realize this, we present an optimization-based TAMP framework where the objectives are learned *functionals* of signed-distance fields (SDFs). The SDFs represent each object in the scene separately, while the functionals defined on top of them induce constraints on possible, physically plausible interactions between the objects within a trajectory optimization problem. The task planning aspect is realized by (discrete) decisions that determine which of those functionals are active at which phase of the planning horizon.

5th Conference on Robot Learning (CoRL 2021), London, UK.

We argue that representing objects as SDFs has multiple advantages. First, an SDF can be seen as an intermediate representation between raw perception like point-clouds or images and full state information. While not the focus of this work, many methods have been developed to learn and obtain SDFs from, e.g., image observations of the scene. Further, SDFs can represent arbitrary, non-convex geometries, which is beneficial, since manipulation problems and physical phenomena often depend on the geometry of the interacting objects. Finally, we show that SDFs are particularly suited for learning and representing models that can later be used within a planning framework effectively. Since our models are functionals of the SDFs, the constraints can take the information about whole objects into account to reason about their geometry and therefore especially the *interaction* between objects. Compared to a representation that only describes the surface of an object like point clouds or occupancy measures, a signed-distance field also provides information about the object at distance. As we experimentally show, this not only leads to models that perform better in their prediction accuracy compared to models learned on top of point-cloud or occupancy object representations, but SDFs also enable the functionals/learned models to have more useful gradients for planning.

In the experiments, we demonstrate the versatility of our approach by tackling two completely different tasks within one framework: On the one hand, a kinematic task where the goal is to hang mugs of different shapes on hooks of different shapes. On the other hand, a pushing scenario where boxes and L-shaped objects should be pushed to different goal regions by pushers of different sizes. In the first case, the model predicts whether the static interaction between SDFs leads to manipulation success, whereas in the latter case, the model predicts the forward dynamics in SDF space based on a history of SDF interactions of two objects. We show that our framework can be used to plan motions that involve multiple push phases. To summarize our main contributions, we propose

- To learn a novel class of kinematic and dynamic models as functionals of SDFs,

- A manipulation planning framework that utilizes these learned functionals as constraints,

- Comparison to other object representations showing the advantages of the SDFs.

## 2 Related Work

### 2.1 Signed Distance Fields as Object Representation

Representing objects or scenes as implicit surfaces [15, 16, 17] or SDFs [18, 19, 20, 21, 22, 23] is an active research topic, due to aforementioned advantages like learning shape completion, non-convex shapes etc. Our focus is not to obtain SDFs from observations in the first place. Conversely, we are interested in what can be done with SDFs in the context of model learning and manipulation planning. There are some works that utilize SDFs within trajectory optimization [24, 25, 26], but without learning or integration into a TAMP framework. While some recent approaches [27, 28, 29] have suggested that grasping of diverse objects can be addressed using implicit functions, we present a manipulation framework that utilizes SDFs for learning and formulating more general models.

### 2.2 Perceptual Models

There is great interest in learning predictive models in perception spaces, especially applied to the problem of pushing. So-called visual foresight approaches [30, 31, 32] aim to predict the evolution of the scene in image space. Our SDF dynamics model is also closely related to perception spaces, but, in comparison, is naturally differentiable. Xu et al. [33] use a voxelized SDF-based representation of the whole scene to predict the motion of an object when an action is applied. Our approach is more structured in the sense that we do not predict the scene flow for actions applied on a single object, but the dynamics of interacting of objects. In [34], the pushing dynamics in keypoints extracted from visual object observations is learned. However, their focus is to utilize the learned model to stabilize a trajectory with MPC. We focus on planning a complex pushing trajectory and not stabilization during execution. SE3 networks [35] learn a forward model that predicts a rigid transformation of an observed point cloud given actions. However, they need ground-truth transformations at training time (we only need SDF observations). Where most of these approaches differ from our approach is that they assume the model to be a function of the observation of a single object or the scene and an action as input. Therefore, these approaches are mostly limited to the same pusher geometry and make the assumption that actions can readily be applied to the object. Our model handles the interaction between objects of different shapes and can plan the contact establishment phase as well. Transporter networks [36] or deep visual reasoning [7] predict manipulation sequences from image spaces. However, no dynamic models are considered in these approaches.

## 2.3 Manipulation Planning (with learned models)

In [37, 38], a manipulation framework based on point cloud observations and manipulation primitives is proposed. Our method plans the complete motions based on learned dynamic models. Sutanto et al. [39] is related to our formulation in the sense that they learn manifolds that are used as constraints in sequential manipulation problems. However, there are no dynamic models or dependencies on the geometry of the involved objects in the learned constraints. You et al. [40] address a hanging task similar to our mug hanging experiment on a more diverse set of object categories. They use a point-cloud-based input representation to predict a hanging pose. Therefore, they need a special neural network for collision avoidance (similar to [41]), while our SDF based representation can handle collisions directly. Further, we learn a manifold of solutions instead of predicting a single hanging configuration. To summarize, what makes our approach unique is that we propose to use SDFs as a common object representation that is closely connected to perception to learn a variety of models that are able to take the interaction of objects into account and can be integrated in an optimization-based motion planning framework due to their differentiability.

## 3 Background on Signed-Distance Fields (SDFs)

Let $\Omega \subset \mathbb{R}^3$ be an object in the 3D Euclidean space. A function $\phi : \mathbb{R}^3 \to \mathbb{R}$, $\phi \in \Phi$ with $\phi(x) = -d(x, \partial\Omega)$ for $x \in \Omega$ and $\phi(x) = d(x, \partial\Omega)$ for $x \in \mathbb{R}^3 \backslash \Omega$ is called a signed-distance field of $\Omega$ in $\mathbb{R}^3$. Here, $d(x, \partial\Omega) = \inf_{x' \in \partial\Omega} \|x - x'\|_2$ and $\partial\Omega$ the boundary of $\Omega$. We assume $\phi$ to be differentiable almost everywhere in $\mathbb{R}^3$. The way $\phi$ is defined ensures that inside the object, $\phi$ attains negative values, on the boundary zeros, and outside positive ones. We denote with the set $\Phi$ the space of all functions $\phi$ that are SDFs for some object.

**Rigid Transformations of SDFs** A central concept in this work is to rigidly transform SDFs in space. This can be realized by transforming the input where the SDF is queried. To simplify the notation, we define a *rigid transformation*, parameterized by $q \in \mathbb{R}^7$ (translation + quaternion),

$$T(q)[\phi](\cdot) := \phi\left(R(q)^T\left(\ \cdot\ - r(q)\right)\right) \tag{1}$$

of an SDF $\phi$, where $R(q) \in \mathbb{R}^{3\times3}$ is a rotation matrix and $r(q) \in \mathbb{R}^3$ the translation vector.

## 4 Manipulation Planning with Signed-Distance Functionals

The core idea of this work is to represent each object $i$ in the scene as a signed-distance field $\phi^i$ in order to learn predictive models as functionals $H$ of these SDFs. Based on the learned functionals, we formulate a trajectory optimization problem where the decision variable is a trajectory of rigid transformations applied on the initial SDFs as they have been observed in the initial scene.

More specifically, through interaction with the environment, we aim to learn functionals of the form $H : \Phi \times \cdots \times \Phi \to \mathbb{R}$ that map multiple SDFs of multiple, possible different objects at possibly different consecutive times to a real number. These are trained in a way that a value of zero implies that the SDFs as input are compatible with what has been learned through interaction with the environment. Otherwise, they should attain a positive value, hence functionals $H$ discriminate correct from incorrect dynamics or desired from undesired manipulations.

The learned functionals then define constraints for the (hybrid) trajectory optimization problem

$$\min_{\substack{q_{0:KT}, q_t \in \mathbb{R}^{7 \cdot n_O} \\ K \in \mathbb{N}, \ s_{1:K}}} \sum_{t=1}^{KT} c\big(q_{t-l:t}, s_{k(t)}\big) \tag{2a}$$

$$\text{s.t.} \quad \forall_{H \in \mathbb{H}(s_{k(t)})} \ : \ H\left(\big(T(q_t^i)[\phi^i]\big)_{(t,i) \in \mathcal{I}_H(s_{k(t)})}\right) = 0 \tag{2b}$$

$$s_{1:K} \in \mathbb{S}(S), \ \ q_0 = 0. \tag{2c}$$

The discrete variable $s_k$ determines which functionals $H$ from the set $\mathbb{H}(s_{k(t)})$ are active at which of the $K \in \mathbb{N}$ phases of the motion ($k(t) = \lfloor t/T \rfloor$). This number of phases is part of the decision problem. The trajectory $q_{0:KT}$ of rigid transformations is discretized in time into $T \in \mathbb{N}$ steps per phase. If $n_O$ is the number of objects in the scene $S$, then $q_t \in \mathbb{R}^{7 \cdot n_O}$, leading to $7 \cdot KTn_O$ continuous variables. Further, $s_k$ selects through the set $\mathcal{I}_H(s_{k(t)})$ the time slice and object index tuples $(t, i)$ that determine the SDFs $\phi^i$, which have been transformed through $q_t^i$, at the times $t$ of the trajectory on which the functional constraints $H$ depends on. This problem formulation is

inspired by LGP [10], but the constraints are replaced by learned functionals of SDFs. The set $\mathbb{S}(S)$ contains all valid sequences $s_{1:K}$ of such discrete variables for the scene $S$. The goal of the manipulation problem is specified through $\mathbb{S}(S)$ by $s_K$ selecting a desired goal functional constraint that has to be fulfilled at the end $q_{KT}$ of the trajectory. Solving (2) therefore involves a tree search over nodes $s_{1:K}$ such that the continuous optimization problem implied by the choice of $s_{1:K}$ at a node of the tree is feasible. The role of $q$ in the optimization problem is not absolute object poses, but rather rigid transformations applied to the SDFs $\phi^i$ that represent the configurations of the objects as observed in the scene initially. With the term $c$, we can include regularizing motion costs. As will be described in sec. 5.1, the forward dynamic model we learn for pushing implies a constraint on the evolution of one object based on the motion of another object. Therefore, we only add motion costs to those degrees of freedom that can be interpreted as being controlled, meaning the motion of the other object. From the perspective of (2), there is no explicit notion of controlled actions.

## 5 Deep Signed-Distance Functionals

This section presents two main types of models we propose. First, a way of learning forward dynamic models that predict the dynamics in SDF space based on the interaction between objects. Second, a kinematic success model that determines whether a static configuration of interacting SDFs leads to manipulation success. All functionals we consider are of the form $H : \Phi \times \cdots \times \Phi \rightarrow \mathbb{R}$, i.e. they only take the SDFs of interacting objects as input, there is no explicit notion of position, orientation, action etc. Therefore, the functionals can be used at arbitrary locations in space.

**Bounding-Box**   To define most of the following functionals and those in sec. 6, we utilize a set $\mathcal{X}$ with the property $\Omega \subset \mathcal{X} \subset \mathbb{R}^3$ for all objects $\Omega$ that are involved. This set should be large enough to cover the relevant workspace of the manipulation problem where the interaction between the objects should occur. A more detailed discussion about the role of $\mathcal{X}$ can be found in sec. 5.3.

### 5.1   Forward Dynamic Models

Generally, a forward model predicts future states/observations of a system given the current or additionally a history of states/observations. In the context of objects being represented solely as SDFs, we propose to learn a forward model $F : \Phi \times \cdots \times \Phi \rightarrow \Phi$ that predicts the SDF of an object $\phi_t^1$ at time step $t$ based on a history of SDF observations of the object $\phi_{t-l:t-1}^1$ until time $t-1$ and the motion of another object $\phi_{t-l:t}^2$ until time $t$. This means $F$ as

$$\phi_t^1(\cdot) = F\big[\phi_{t-l:t-1}^1, \phi_{t-l:t}^2\big](\cdot) \tag{3}$$

is an SDF itself that can be queried in $\mathbb{R}^3$. Interactions between more than two objects are possible, but we focus on pair-interactions in the present work. If $l = 1$, $F$ is a quasi-static model. Internally, $F$ can be defined to either directly predict the SDF $\phi_t^1$ as in (3) or the flow

$$\phi_t^1(\cdot) = \phi_{t-1}^1(\cdot) + F_{\text{flow}}\big[\phi_{t-l:t-1}^1, \phi_{t-l:t}^2\big](\cdot) \tag{4}$$

from $\phi_{t-1}^1$ to $\phi_t^1$. In both cases, the functional $H$ for planning is then naturally defined as

$$H_F\big(\phi_{t-l:t}^1, \phi_{t-l:t}^2\big) = \int_{\mathcal{X}} \left(\phi_t^1(x) - F\big[\phi_{t-l:t-1}^1, \phi_{t-l:t}^2\big](x)\right)^2 \mathrm{d}x. \tag{5}$$

For a perfect model $F$, this functional $H_F$ attains a zero value if and only if the evolution of $\phi_{t-l:t}^1$ and $\phi_{t-l:t}^2$ is compatible with the underlying physical process in the space $\mathcal{X}$. Therefore, the loss function to train $F$ is also (5) for a dataset $D = \big\{\big(\phi_{0:l}^1, \phi_{0:l}^2\big)_i\big\}_{i=1}^n$ of such consecutive SDF motions of the two objects. Since $F$ takes as input the complete SDFs of the objects and not just values like the distance between objects and their contact point locations, it can learn to reason not only about these quantities, but also the contact geometry, relative object movements, center of mass and inertial parameters (assuming an equal density of the objects), all of which are necessary quantities to represent the dynamics. This way, $F$ inherently takes the geometry of the objects into account. Note that usually, forward models are understood in terms of a function that maps the current state (history) and an (abstract) action $a$ to the next state. For SDFs, this would mean a model of the form $\phi_t^1 = F[\phi_{t-l:t-1}^1, a_{t-1}]$. In our case, however, there is no notion of an abstract action, instead, our formulation learns a generic model of the interaction between two objects, where the motion of one object ($\phi^2$) influences the other ($\phi^1$). Therefore, while the transformation applied to $\phi^2$ can be interpreted as an action, the model has no action as input and hence can deal with different geometries of $\phi^2$, which is not possible in case of an abstract action without $\phi^2$ also being an input.

## 5.2 Kinematic Success Models

Many tasks in manipulation planning can be specified in terms of static success models instead of a full forward dynamics model. We call a model that predicts whether a configuration of potentially multiple SDFs at the same time slice leads to manipulation success a kinematic success model. Assume through interaction with the environment, a dataset $D = \left\{ (\phi^j)_{j \in \mathcal{I}_i}, y^i \right\}_{i=1}^{n}$ of SDFs representing $|\mathcal{I}_i|$ many objects has been obtained with $y^i = 1$ indicating that the configuration of SDFs leads to manipulation success, $y^i = 0$ to failure. Then learning $H$ is similar to a classification problem, where $H\left( (\phi^i)_{i \in \mathcal{I}} \right) = 0$ implies success prediction. This way, $H$ can model a manifold of feasible configurations and not only a single solution. See sec. F for details (loss function etc.).

## 5.3 Learning Functionals with Neural Networks

So far, we have not discussed how functionals of the form $H : \Phi \times \cdots \times \Phi \to \mathbb{R}$ can be learned or even queried in the first place with usual function approximators like neural networks, since, in general, the neural network would have to take functions as infinite dimensional objects as input. To approximate this, we choose in this work the straight-forward approach by evaluating $\phi \in \Phi$ on a discretized version of the set $\mathcal{X}$, denoted by $\mathcal{X}_h$. As discussed previously, the set $\mathcal{X}$ should cover the relevant region of the workspace where the interaction between the objects takes place. We specifically do not assume $\mathcal{X}$ to be aligned or perfectly centered with the objects that are involved. This way, the dynamics model from sec. 5.1 can be realized by

$$F\left[\phi_{t-l:t-1}^1, \phi_{t-l:t}^2\right](x) \approx F_\theta(\phi_{t-l:t-1}^1(\mathcal{X}_h), \phi_{t-l:t}^2(\mathcal{X}_h), x) \tag{6}$$

with $F_\theta$ being usual neural network architectures. During training, the integral in (5) is approximated over the same discretized $\mathcal{X}_h$ for simplicity. Hence, the dataset to train $F$ can contain the SDF observations at the grid points of $\mathcal{X}_h$ only. However, $F_\theta$ still approximates an SDF which can be queried at arbitrary $x \in \mathbb{R}^3$ and does not only predict the values on the grid points. For general functionals $H$, the evaluation is analogous, i.e. $H\left( (\phi^i)_{i \in \mathcal{I}} \right) \approx H_\theta \left( (\phi^i(\mathcal{X}_h))_{i \in \mathcal{I}} \right)$. Technically, $\mathcal{X}_h \subset \mathbb{R}^{d \times h \times w}$ is a regular grid which allows us to encode $\phi(\mathcal{X}_h)$ using 2D or 3D convolutions. In contrast to an occupancy grid, the evaluation of $\phi(\mathcal{X}_h)$ contains more information about the object than whether there is an object at the grid point or not. Note that the differentiabilty of $H\left( (T(q^i)[\phi^i(\mathcal{X}_h)])_{i \in \mathcal{I}} \right)$ with respect to $q^i$ is maintained, which is another advantage of representing such models as functionals of SDF functions evaluated on a grid instead of static values on a grid. During training, it is sufficient to only have the SDF values evaluated on a gird, no other information like actions or velocity/pose estimations are needed.

## 6 Task Constraint Functionals

Here we present analytical functionals of SDFs that are useful to specify goals of a manipulation problem or other task aspects. These functionals are general as a direct consequence of our object representations being SDFs. Therefore, there is no advantage or need to learn these given the SDFs.

### 6.1 Pair-Collision between Objects

Collision avoidance is an inherent part of many task specifications. Given two SDFs $\phi_1, \phi_2$, we can measure whether they are in collision via their overlap integral

$$H_{\text{coll}}(\phi_1, \phi_2) = \int_\mathcal{X} [\phi_1(x) < 0]\, [\phi_2(x) < 0]\, \mathrm{d}x. \tag{7}$$

The indicator bracket $[\cdot]$ means $[P] = 1$ if $P$ is true, otherwise $[P] = 0$. The integral in (7) integrates over the space where both SDFs are negative at the same time, which is only the case if the two objects overlap, hence are in collision. The gradients of (7) are smoothed using the sigmoid function $\sigma(z) = \frac{1}{1+\exp(-z)}$, i.e. $[\phi_{1,2}(x) < 0] = \sigma\left(-a\phi_{1,2}(x)\right)$ with a parameter $a > 0$.

### 6.2 Goal Region

If part of the task specification is that an object $\phi_1$ is fully contained inside the boundary of another object $\phi_g$, called the goal region, then a similar integral as for the pair-collision can be utilized

$$H_{\text{g}}(\phi_1, \phi_g) = \int_\mathcal{X} [\phi_1(x) < 0]\, [\phi_g(x) > 0]\, \mathrm{d}x \approx \int_\mathcal{X} \sigma\left(-a\phi_1(x)\right) \sigma\left(a\phi_g(x)\right) \mathrm{d}x. \tag{8}$$

Here, points outside of the goal region that are inside the object count towards the integral.

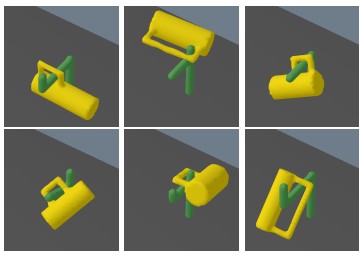

Figure 1: Found solution configurations by the optimizer using the learned model.

| | solution found | success rate | **total scenes solved** |
|---|---|---|---|
| **SDF opt. + sampling** | 98.7% | 88.5% | **87.3%** |
| SDF opt. only | 51.3% | 93.5% | 47.3% |
| SDF sampling only $\kappa_1$ | 83.8% | 82.3% | 68.9% |
| occupancy $\psi$, sampling $\kappa_3$ | 100.0% | 34.0% | 34.0% |
| pointnet++, sampling $\kappa_1$ | 100.0% | 82.0% | 52.0% |

Table 1: Success rates of mug hanging experiment. Total scenes solved means the percentage of scenes in the evaluation dataset for which a solution was found that is not in collision and is stable when dropped. Only best results for each object representation shown. Full results see Tab. 3 and sec. A.5.

## 6.3 Establishing Contact between Objects

Establishing and maintaining contact between objects is central for many manipulation tasks. One way to model that the distance between two objects $\phi_1$ and $\phi_2$ should be zero is via the functional

$$H_{\text{PoC}}(\phi_1, \phi_2) = \min_{p \in \mathcal{X}} |\phi_1(p)| + |\phi_2(p)|. \tag{9}$$

# 7 Experiments

## 7.1 Mug-Hanging: Kinematic Success Model

In this experiment, we want to find rigid transformations applied on observed mugs of different shapes in a scene to hang them stably on hooks of different types. The functional $H_{\text{hang}}$ is therefore a kinematic success model that takes the SDFs of the mug and the hook as input. To generate data to learn $H_{\text{hang}}$, we randomly sample scenes of different mug and hook shapes (1600 scenes for training, 400 for testing and 150 for evaluation). See Fig. 15 for examples of mugs and hooks in the evaluation data. Then we sample for each scene in the training and test data the position and orientation of the mug uniformly in the bounding box $\mathcal{X}$ until at least one successful configuration has been obtained where the mug does not fall on the ground when being dropped from the sampled configuration while at the same time not being in collision with the hook. We use Bullet [42] to simulate the dropping. In total, 20 configurations per scene are generated. Since sampling a successful configuration is a rare event, for the majority of the scenes, only one successful and 19 failure configurations are contained in the training and test data, making learning challenging. Another challenge of this task is that the model has to reason about both the hook and mug geometry jointly. Formulating an analytical model, e.g. on a mesh-based object representation, to model this constraint is non-trivial.

### 7.1.1 Performance with Optimization

Fig. 1 shows solution configurations found by our model $H_{\text{hang}}$ as an optimization objective. Interestingly, the solutions not always contain the intuitive solution, but also ones where other parts of the hook are being utilized (middle column in Fig. 1). The optimization problem (2) to solve this mug hanging problem has two objectives, the learned kinematic success functional $H_{\text{hang}}$ and the pair-collision $H_{\text{coll}}$ from sec. 6.1. While in principle $H_{\text{hang}}$ also learns to avoid collisions, we found that the robustness in avoiding collisions increases when including $H_{\text{coll}}$. The learned functional $H_{\text{hang}}$ is, in general, non-convex in the rigid transformation $q$ of the mug. Therefore, we observed that gradient-based optimization is not sufficient for the optimizer to find a feasible solution, i.e. where $H_{\text{hang}}$ predicts zero, in every instance [43]. To overcome this issue, we restart the optimization procedure up to 20 times with a randomly sampled initial guess of the mug in $\mathcal{X}$. Fig. 16 shows an example of a sampled initial configuration from which the optimizer is started (left), then the optimized configuration (middle) and finally, the configuration after simulation. Tab. 1 shows the success rates on the evaluation scenes. As one can see, for the proposed approach where objects are represented as SDFs using optimization and sampling, in 98.7% of the evaluation scenes, a solution is found where $H_{\text{hang}}$ predicts success and no collision is violated (first column). Out of these, 88.5% are stable configurations (checked by simulation) and the optimized configuration of the mug is collision free with the hook, leading to 87.3% total solved scenes (last column). When the optimization is run only once (second row), then only in 51.3% of the cases it converges to a feasible solution.

### 7.1.2 Comparison to Sampling, Point-Cloud and Occupancy Measure Representations

This section shows that learning a kinematic success model based on the proposed SDF object representation outperforms other representations (point-clouds and occupancy measures) and further highlights the advantage of the models learned with SDFs providing useful gradients by comparing it to sampling without optimization. For full results, refer to sec. A and Tab. 3. The sampling

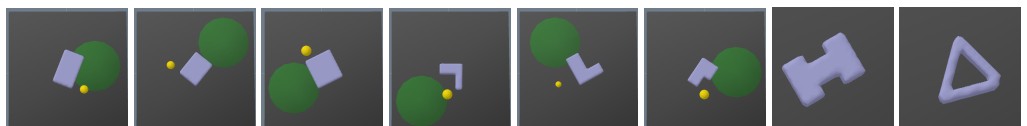

Figure 2: Different pushing scenarios in evaluation dataset. Yellow is the pusher, green the goal region and light blue the object. The two images from the right show out-of-distribution generalization shapes.

approach draws relative transformations of the mugs uniformly in $\mathcal{X}$ until the evaluation with the learned $H_{\text{hang}}$ and the collision functional $H_{\text{coll}}$ predicts a successful and collision free configuration with a threshold $\kappa$. As one can see in Tab. 1 and Tab. 3, our proposed approach has a significantly higher performance (87.3%) compared to the best threshold for pure sampling with SDF (68.9%) and the best of the other object representations (34% with occupancy measure, 52% with point-cloud).

## 7.2 Pushing Objects on a Table: Dynamic Model

In this experiment, we consider the task of pushing boxes and L-shaped objects of different dimensions with a spherical pusher of different radii into a goal region $\phi_g$ on a table. Fig. 2 visualizes typical objects, pushers and goal regions. The goal region has to be large enough that all possible objects fit. We again use Bullet as a simulator to generate data to train a dynamics model of the from described in sec. 5.1 with $l = 1$, i.e. $H_F$ is a function of four SDFs $\phi_t^1, \phi_{t-1}^1$ (object) and $\phi_t^2, \phi_{t-1}^2$ (pusher). In total, 14975 different scenes (including shapes and initial configuration) are sampled where random push actions biased roughly towards the object center are applied until the object leaves the table. Since the dynamics and interaction of the objects in this scenario can be described in the 2D plane, the 3D signed distance functions of the objects are evaluated in the 2D set $\mathcal{X}_h \in \mathbb{R}^{140 \times 140}$ only. Therefore, the model $F$ predicts the dynamics of $\phi^1$ in this 2D projection.

### 7.2.1 Forward Prediction Error

Tab. 2 shows the one-step prediction error on the evaluation dataset for the flow model $F_{\text{flow}}$ (4) and the direct SDF prediction $F$ (3). The way we utilize the model within the trajectory optimization problem never asks for predictions more than one step into the future. We

|  | contact phase | no contact phase |
|---|---|---|
| $F_{\text{flow}}$ | $3.4 \pm 1.6$ | $1.4 \pm 1.8$ |
| $F$ | $5.8 \pm 1.7$ | $5.2 \pm 1.6$ |
| $\phi_t^1 = \phi_{t-1}^1$ | $10.8 \pm 3.4$ | $0$ |

Table 2: RMSE [mm] on evaluation dataset.

train one single dynamics model for both box and L-shaped objects and different pushers. The prediction error is the RMSE of predicting the correct SDF values in $\mathcal{X}_h$. The last row shows the error if the model would simply predict the next state as the last state of the object. As one can see, $F_{\text{flow}}$ achieves a lower error than $F$. This is due to $F$ having to predict the complete SDF, while $F_{\text{flow}}$ only the flow. In phases of the motion where there is no contact between the object and the pusher, both models $F_{\text{flow}}$ and $F$ have to learn that the object should not move (and $F$ has to predict the complete SDF in this case as well), which is also non-trivial, but they accomplish this with low error.

### 7.2.2 Comparison to other Object Representations (Point-Cloud and Occupancy Measure)

In sec. A.6.1 and sec. A.6.2 we present and explain a comparison of the forward prediction error between models learned with object representations being SDFs, occupancy measures and point-clouds. As shown in Tab. 4 and Tab. 5, models learned with the SDF representation outperform models based on point-clouds and occupancy measures in their predictive performance. Further, in Tab. 5 and sec. A.4, we also show that one can learn image conditioned SDFs and dynamic models on top of the learned SDF simultaneously with no noticeable performance degradation.

### 7.2.3 Planning with the Learned Model and Execution Performance

Having learned the pushing dynamics prediction model, we now utilize it within (2) to solve the task of pushing the object into the goal region. There are four constraints. First, the dynamics model $H_F$ and, second, the goal region $H_g$. While this seems to be enough to specify the problem fully, we add two additional constraints, $H_{\text{coll}}$ and $H_{\text{PoC}}$. The discrete variable $s_k$ of (2) decides whether there are one or two push phases. Only in a push phase, $H_{\text{PoC}}$ is active. $H_{\text{coll}}$ is always active. Similarly to the mug hanging experiment, local minima are a core issue as well. Therefore, we initialize the pusher position at phase 1 or 2 on a set of 4 different points around the object. These 4 points around the object are always the same in all scenarios, no matter of the size, shape or orientation of the object. Compared to other approaches where the action space has to be chosen much more carefully, we believe that this is a rather weak prior. The initialization also does not start from contact with the object or similar, because our problem contains the challenge of contact establishment and possible breakage to push from a different side to achieve the goal. Therefore, due to this initialization, there

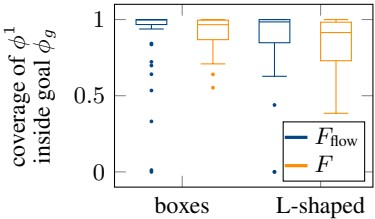

Figure 3: Pushing performance on evaluation scenes in terms of the amount of $\phi^1$ that is inside of the goal region at the end of the execution. A value of 1 means that the object is fully contained in the goal region.

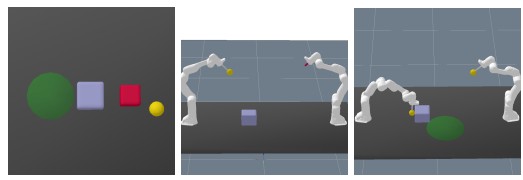

Figure 4: Generalization experiments for pushing scenario. Left: interaction between three objects to solve the task. Middle: the goal is that the right robot arm touches the object. Right: the two robots have to collaborate to push the object into the goal region, which would not be possible with one arm alone.

are 20 different optimization problems we solve for each scene (4 for one pushing phase, $4^2$ for two pushing phases). To evaluate the performance, we execute the planning result with the least constraint violation and cost *open-loop* in the simulator. Despite the fact that pushing is unstable over long-horizons, our proposed approach achieves a high performance. As shown in Fig. 3, which plots the amount of the object that is inside of the goal region at the end of the execution, using the learned $F_{\text{flow}}$, the approach achieves 99.7% (median) coverage of the object inside the goal region on box pushing and 98.4% (median) on the L-shaped objects (50 evaluation scenes each). Please note that, although the goal region for small objects seems large, the optimizer usually moves the object until it is just barely inside the goal region and not any further. Therefore, even very small deviations during the open-loop execution already lead to some parts sticking out. For the larger objects in the evaluation scenes, the goal region is barely large enough. With the direct $F$, the performance is a bit worse, but still high (median 96.6% for boxes, 91.5% for L-shaped objects).

Sec. B demonstrates that our proposed SDF framework outperforms an approach where the optimization problem is formulated for objects represented as meshes and the analytic dynamic model from [13] for the pushing dynamics. Refer to Fig. 6 for quantitative results. Finally, in sec. A.6.3 and Fig. 5 we investigate the importance of models learned on top of SDF representations providing useful gradients for planning in comparison to models with point-clouds and occupancy measures.

### 7.2.4 Ablation Study

Sec. D presents an ablation study regarding the importance of the additional objectives $H_{\text{coll}}$ and $H_{\text{PoC}}$ in the optimization problem. Results in Fig. 11 show that while it is possible to solve the scenes without them, the performance greatly increases if they are part of the problem formulation.

### 7.2.5 Generalization to Out-of-Distribution Shapes, Multiple Objects, and Robots

In sec. C.1, we demonstrate that the framework and the learned model generalizes to shapes beyond the training distribution. See Fig. 2 and Fig. 7 for those shapes. Quantitative results presented in Tab. 6 and Fig. 8 indicate that the model achieves both high prediction accuracy and high performance when used for planning. Furthermore, as seen in Tab. 6, a model learned with SDFs generalizes significantly better than with point-clouds, also relative to the results obtained on-distribution. We further show in sec. C.3 and sec. C.2 that the framework is capable of generalizing to scenes that contain obstacles (Fig. 10) and a scenario where three objects interact in order to solve the task (Fig. 9 and Fig. 4). Finally, in sec. E we demonstrate multiple scenarios where the learned pushing dynamics model is embedded into a scene that contains robots. All these generalization experiments require no change in the methodology or to learn a new model, showing the generality and versatility of our proposed framework. For more details, refer to the respective sections in the appendix.

## 8 Conclusion

In this work, we have shown that the constraints of a trajectory optimization problem for solving manipulation problems can be formulated in terms of learned functionals of SDFs only. SDFs can serve as a *common* object representation across completely different tasks. The functionals can naturally model the interaction between objects of arbitrary shapes and can be learned directly from SDF observations, which is closely connected to perception. We have shown that learning models on top of SDFs outperform other object representations like point-clouds and occupancy measures both in terms of prediction accuracy and the ability to plan. The greatest challenge of our framework are local minima of the resulting trajectory optimization problem. While sampling strategies for initial guesses can mitigate this to some extend, it is an issue, which is not unique to our approach, but many nonlinear trajectory optimization formulations. While we have considered rigid objects in this work only, we believe that the proposed approach can be extended to deformables as well.

**Acknowledgments**

Danny Driess thanks the International Max-Planck Research School for Intelligent Systems (IMPRS-IS) for the support. This research has been supported by the German Research Foundation (DFG) under Germany's Excellence Strategy – EXC 2002/1–390523135 "Science of Intelligence". The authors thank the anonymous reviewers for their comments.

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
