# OpenReview forum: "Learning Models as Functionals of Signed-Distance Fields for Manipulation Planning"
_robot-learning.org/CoRL/2021/Conference — CoRL2021 Poster_

### Official Review · Reviewer_NU21 · 2021-07-23

**Originality:** Fair
**Technical Quality:** Good
**Clarity Of Presentation:** Very Good
**Impact:** 3

**Recommendation:**

Weak Accept: I recommend accepting the paper, but will not argue for my recommendation if the majority of other reviewers have a different opinion.

**Summary:**

This paper aims to solve the TAMP problem for tasks that require kinematic and dynamic reasoning beyond the capabilities of analytical constraint modelling. Hence they propose to learn these models using SDF representations of different objects in the scene. The authors claim that this allows the model to learn various geometric and dynamic interactions implicitly which are necessary to solve the manipulation problems of interest.


**Issues:**

- Authors claim to propose a TAMP framework, however, it seems more suitable to say that they propose a learning-based approach to evaluating task constraints which facilitates task planning.
- While promising results have been achieved for the demonstrated tasks, there remains the technical challenge of how this method would scale to other scenarios. For example, for the proposed forward dynamics model, it seems non-trivial to have more than two interacting objects. Furthermore, the complexity of computing the interactions will increase non-linearly. Additionally, the training process seems quite lengthy and might be prohibitively slow for multiple objects, scenes and tasks. This could be further compounded depending on the size and resolution of the SDF.
- Dealing with the non-convexity of the problem via random restarts is unsatisfying and could be problematic depending on the complexity of the problem.
- Some grammatical errors.
- Regarding the video, there is a lot of repeated experiments that seem similar and have no apparent added value.
- Furthermore, it would be helpful to show the significance of your results by showing a comparison of your method alongside the failure cases of other methods.


**Reviewer Expertise:**

Good: General knowledge of the area

**Strengths And Weaknesses:**

Strengths

The authors have a good grasp on the current learning-based approaches and gaps in the literature which helps motivate their problem.
The authors provide some interesting task constraint formulations that can be used for learning and solving two particular manipulation tasks with SDFs as inputs.
The method is described well and the formulation is sound.
The choice of SDF representations to capture models and formulate the optimisation problem is key in the framework.

Weaknesses

While the method is sound and well-motivated, there is a lack of experimental evidence to justify the need for their method over any of the many existing ones for solving such problems.



**Summary Of Recommendation:**

The proposed formulation is interesting and well-motivated, however, due to weak experimental evaluation coupled with the incremental nature of the contribution it is difficult to justify publication.

---

> ### Author Response · Authors · 2021-08-31
> **Thanks for your review, we have added many comparisons to other object representations**
>
> Thanks a lot for your comments, which helped us a lot to clarify and strengthen the paper.
>
> > While the method is sound and well-motivated, there is a lack of experimental evidence to justify the need for their method over any of the many existing ones for solving such problems.
>
> We agree that the initial submission contained too little comparisons (both experimentally and explanatory) to other methods.
> We therefore added extensive comparisons and discussions to models learned on point-cloud representations and implicit occupancy measures (with the latter already part of the appendix in the initial submission, but had not been mentioned in the main text).
> Results show that our proposed approach outperforms other methods both in terms of model learning and planning capabilities.
>
> Further, we added more complex scenarios as an application of the formalism.
> These include integration of robots into the scene, three interacting objects and obstacles in the scene.
> This required no relearning or changes to the framework, highlighting the versatility of the method, which is not possible with related approaches.
> We are not aware of any other method that can address these tasks in the same generality.
>
>
> > Authors claim to propose a TAMP framework, however, it seems more suitable to say that they propose a learning-based approach to evaluating task constraints which facilitates task planning.
>
> Thanks for pointing this out. We would like to clarify that the proposed framework is capable of joint task and motion planning for multi-step/multi-phase manipulation problems (using learned models). Due to your comments, we have added many more demonstrations showing TAMP like scenarios, including two robots, three interacting objects and obstacles. We would like to stress that the formalism remained the same in these experiments and no relearning of the functionals is necessary.
>
>
> >While promising results have been achieved for the demonstrated tasks, there remains the technical challenge of how this method would scale to other scenarios. For example, for the proposed forward dynamics model, it seems non-trivial to have more than two interacting objects. Furthermore, the complexity of computing the interactions will increase non-linearly. Additionally, the training process seems quite lengthy and might be prohibitively slow for multiple objects, scenes and tasks. This could be further compounded depending on the size and resolution of the SDF.
>
> Since our learned models are object centric, they can be applied in new environments that for example contain other objects without the need to relearn. To demonstrate this, we have added an experiment containing an obstacle.
> Additionally, we have added another experiment where object 1 pushes object 2 which pushes object 3 to the goal location. Although 3 objects interact, we do not have to learn a new model. Thanks for making the point.
>
> Having one object pushing two objects at the same time directly (e.g. one large box pushing two smaller boxes in front of it) would also be possible. What would not work with the current trained push model is to have two objects pushing one other object at the same time. However, our general problem formulation would be able to deal with that case as well, but would require training a new model that contains such situations (and the model would have to take the motion of the two pushing objects as input).
>
> The fact that object interactions scale non-linearly with the number of objects involved is caused by physics and not unique to our formulation. Of course, typical tricks could be applied to our problem formulation such as only considering object interactions when the objects are reasonably close to each other, which greatly reduces this scalability problem.

---

> > ### Author Response · Authors · 2021-08-31
> > **continue**
> >
> > >Dealing with the non-convexity of the problem via random restarts is unsatisfying and could be problematic depending on the complexity of the problem.
> >
> > When solving hard TAMP problems, we believe there are no generic methods available that can address non-convexity in a satisfying way.
> > We are aware of three typical ways of dealing with that issue. One is completely sampling based, which usually requires a lot of samples or carefully engineered task-specific sampling distributions. Another is to decompose the problem into convex subproblems, which is not only challenging for general problems to do in the first place, but usually implies a huge combinatorial complexity (although we are sympathetic to this approach and the LGP formulation our approach is based on is inspired from this). Lastly, one can rely on nonlinear optimization, which has the issue of potentially converging to infeasible local optima. However, when combined with sampling, nonlinear optimization can achieve good results. We totally agree that the non-convexity is a challenge (if not the biggest of robot manipulation planning in general) and therefore, we evaluated it in our experiments. As we have shown, for the pushing scenario, no random restarts were needed, since the four rough initializations around the object (always the same initializations, independent from the concrete object geometry or its rotation) were sufficient. In this case, the discrete decisions of the initialization solved the problem. For the hanging scenario, there is no clear discrete structure. Therefore, we sample the initializations in the 6D relative transformation space. Combined with optimization, this works robustly. As we have shown experimentally, sampling alone is much worse than our combined approach of sampling and optimization. Further, the number of restarts was limited to 20.
> >
> > >Regarding the video, there is a lot of repeated experiments that seem similar and have no apparent added value.
> >
> > Our intention was to show that the method does not only address a single scene setup with a single object/pusher geometry or same pusher initialization, but instead a large variety of scene parameters. Further, one can see the different behaviors the system creates as well as some failure cases. We agree that it, when watching it the first time, seems repetitive. We tried to improve on that point. Furthermore, we have added many more examples showing what the system is capable of in the video.
> >
> > >Furthermore, it would be helpful to show the significance of your results by showing a comparison of your method alongside the failure cases of other methods.
> >
> > As mentioned above in the response to the AC, we totally agree that we have not put enough emphasis in the initial submission to comparisons to other methods. Therefore, we have not only added several quantitative comparisons to other methods, but also discussed them and made the comparisons more prominent in the main text (the comparison to occupancy measures had already been in the initial submission in the appendix, but was not mentioned in the main text). Due to severe space limitations, the results of the comparisons and details are presented mainly in the appendix.
> >
> > Results show that learning models on top of objects being represented as SDFs outperforms all other methods with respect to prediction performance in case of the pushing scenario and success rate in case of the mug hanging scenario.
> >
> > Regarding failure cases of other methods, in case of the pushing scenario, neither models learned from point-cloud representations nor from occupancy measures provide gradients that are useful for planning, even in completely simplified scenarios where the object should just be pushed straight and the contact between the pusher and the object had already been established in the initialization.
> > Therefore, the point-cloud and occupancy representation are not suitable for planning (especially not for the more complex scenarios we consider), while the SDF representation provides informative gradients, enabling our framework to work.
> >
> > For the hanging scenario, a model based on the pointnet neural network architecture provided a little bit more useful gradients. However, it is less clear how to do differentiable collision checks between point-clouds.
> > When using our SDFs for collision checks, but the pointnet for the success model, the method indeed works to some extent, but significantly inferior to our proposed approach.
> >
> > Finally, we also considered a fully analytical approach for the pushing scenario, where it revealed inferior performance compared our proposed approach.
> > We would also like to mention that for the pushing scenario it was possible to write down an analytic model for the pushing dynamics based on rigid body equations, but for the hanging tasks, it is unclear how one could define an analytic success model.

---

### Official Review · Reviewer_SZWq · 2021-07-31

**Originality:** Good
**Technical Quality:** Good
**Clarity Of Presentation:** Good
**Impact:** 3

**Recommendation:**

Weak Accept: I recommend accepting the paper, but will not argue for my recommendation if the majority of other reviewers have a different opinion.

**Summary:**

This paper modifies an existing formulation of task and motion planning (the Logic Geometric Programming framework) to incorporate constraints that are based on implicit surface representations of rigid-body objects. The paper formalizes what a dynamics model over Signed Distance Fields means, as well as other typical constraints present in TAMP problems, for example contact constraints, goal region constraints, as well as pairwise collision constraints.

The motivating argument that the paper puts forth for TAMP with implicit surface representations is that traditional object geometries used in TAMP problems (eg. simple parametric shapes, or even meshes) are limited. In fact the paper claims that "it is unclear how these models can be grounded from sensor information." The paper mentions that the main advantages of SDFs are their ability to represent nonconvex objects, and that it is easier for them to incorporate sensor data, particularly depth data or pointclouds. The main underlying SDF representation assumed in the proposed network are neural networks trained from sampled points inside each object.

I'd say the main contribution of the paper is an explicit formulation of TAMP with implicit surface constraints. There have been other related works that have followed similar approaches in the past using trajectory optimization, but not full TAMP.

**Issues:**

I would very much hope to see the weaknesses I listed as major addressed in the revised version. If I see convincing changes regarding the major issues, I am willing to update my recommendation to weak accept.

**Reviewer Expertise:**

Good: General knowledge of the area

**Strengths And Weaknesses:**

Strengths

S1. This is an interesting idea. Implicit surface representations are definitely well studied in graphics and animation, and the robotics community should pay more attention to them when it comes to motion planning.

S2. The paper is generally clearly written, with the exception of a few sections that I mention below.

Weaknesses (* indicates major)

W1*. The reason presented as the motivation for studying SDFs over traditional parametric shapes used in TAMP is a bit misleading. It is unclear why robot perception would make learning SDFs any easier than parametric shapes or meshes. This is especially true when SDFs are represented with neural networks, which are generally much more difficult to update in an incremental/continual setting than shapes or mesh representations. Also, meshes make it easier to represent objects at different levels of fidelity according to the relevance of the object to the task at hand. I think the writing in the introduction should be updated to reflect more practical reasons why SDFs could be preferable.

W2. It is not clear how scalable this method is. The simulation experiments show very limited planning scenarios. How does this method scale when the number and volume of objects grows?

W3*. The paper seems to be unaware of important prior work from the robotics community on this topic: https://motion.cs.illinois.edu/papers/WAFR2018-Hauser-SemiInfinite.pdf
https://motion.cs.illinois.edu/papers/ICRA2021_Zhang_SIPCC.pdf
These papers should not only be cited, but studied carefully for these types of problems. Other related papers include:
https://www.graphicon.ru/html/2003/Proceedings/Technical/paper495.pdf
https://mmacklin.com/sdfcontact.pdf

W4*. The TAMP problem formulation (2a-c) comes from LGP, but as far as I can understand, the paper resorts to gradient based optimization with multiple restarts, without making use of the discrete tree search that LGP makes use of for the symbolic state. This seems prone to local minima. Why was this choice made?

W5. The presentation of 2a-c is too notation-heavy. Please simplify it for readability.

W6*. SDFs are clearly promising for the dynamics prediction part of the problem. This is evident from the pushing experiments. However, they introduce an inner optimization problem when establishing contact between objects (eqn 10), so it is not clear why the added complexity and cost of including an optimization problem as a constraint is worth it, compared to other representations.

**Summary Of Recommendation:**

I am recommending a weak reject. I think the problem is interesting and difficult, and more people in the robotics community should be looking at it. I think this paper needs a lot more work to provide a convincing solution to the problem, and the current experiments are too simple for the paper to have any impact, but it is enjoyable to read. It would be appreciated if more challenging planning environments could be included, possibly with more objects.


POST-REBUTTAL UPDATE:
--------------------------------------

Thank you to the authors for addressing my concerns in such detail. I am upgrading my recommendation to weak accept. Best of luck!

---

> ### Author Response · Authors · 2021-08-31
> **Thanks a lot for you helpful comments!**
>
> >I'd say the main contribution of the paper is an explicit formulation of TAMP with implicit surface constraints. There have been other related works that have followed similar approaches in the past using trajectory optimization, but not full TAMP.
>
> We would like to add that the contribution of the paper is not only the formulation of TAMP with implicit surface constraints, but also to learn a novel class of models based on these SDFs, which turn out to have informative gradients for motion planning. Further, we are not aware of any previous work showing planning on this level of task complexity with learned models in observation spaces.
>
> >W1*. The reason presented as the motivation for studying SDFs over traditional parametric shapes used in TAMP is a bit misleading. It is unclear why robot perception would make learning SDFs any easier than parametric shapes or meshes. This is especially true when SDFs are represented with neural networks, which are generally much more difficult to update in an incremental/continual setting than shapes or mesh representations. Also, meshes make it easier to represent objects at different levels of fidelity according to the relevance of the object to the task at hand. I think the writing in the introduction should be updated to reflect more practical reasons why SDFs could be preferable.
>
> Obtaining SDFs in the first place is not the focus of this work, since many approaches have been proposed to do so.
> Instead, we argue that SDFs are suitable for both learning models and planning with them in trajectory optimization.
> Despite, we have added an experiment where the SDF is also trained simultaneously from an image observation. Results show only an insignificant drop in performance. There is substantial evidence from the computer vision community in the last two years that SDFs can be obtained for example from image observations of the objects, while providing shape completion and the ability to incorporate multiple views.
> Furthermore, there are also more traditional approaches that estimate SDFs from point-clouds such as TSDFs.
>
>
> >W2. It is not clear how scalable this method is. The simulation experiments show very limited planning scenarios. How does this method scale when the number and volume of objects grows?
>
> Thanks for raising this point, which lead us to make another experiment.
> We agree that TAMP approaches often consider scenes with many objects. However, usually very simple geometries and other simplifying assumptions are made to achieve the scalability there.
> We would argue that our approach in principle is scalable to increasing numbers of objects.
> Compared to other learning approaches that try to model the evolution of the whole scene (e.g. "C. Finn, S. Levine, Deep Visual Foresight for Planning Robot Motion, ICRA 2017", "Z. Xu et al., Learning 3d dynamic scene representations for robot manipulation. CoRL 2020."), the models we learn are object centric.
> Therefore, one can apply these models in scenes that contain more objects.
> Indeed, due to your request, we show a pushing planning scenario that contains an additional obstacle.
> There is no need to relearn the dynamics model.
> Furthermore, we also show that the model that has been trained on the interaction of two objects can be used in a scenario where three objects interact.
> From this perspective, we believe that our approach in the long-term is more scalable to multiple objects compared to other approaches that model the complete scene in one model, where it is questionable how to use it in a completely different scene. One has to mention that a big challenge when incorporating many objects in the scene is the increased non-convexity due to collision avoidance. This, however, is not unique to our approach, but (gradient-based) trajectory optimization in such scenarios in general.
>
> >W3*. The paper seems to be unaware of important prior work from the robotics community on this topic [...]: These papers should not only be cited, but studied carefully for these types of problems. Other related papers include [...]
>
> Thanks for pointing us to these papers. We have added them in the related work section.

---

> > ### Author Response · Authors · 2021-08-31
> > **continue**
> >
> > >W4*. The TAMP problem formulation (2a-c) comes from LGP, but as far as I can understand, the paper resorts to gradient based optimization with multiple restarts, without making use of the discrete tree search that LGP makes use of for the symbolic state. This seems prone to local minima. Why was this choice made?
> >
> > Thanks for pointing this out.
> > The problem formulation (2) is an LGP (where the constraints are now (learned) functionals of SDFs) and indeed makes use of the discrete tree search for the symbolic state.
> > For the pushing scenario, the discrete decisions determine the number of push phases (e.g. a single or two push phases where the pusher is constrained to maintain contact with the object in each phase).
> > In addition, an other discrete decision determines one of four initializations around the object to address the local minima issue of moving around the object to push from different sides (as explained in the paper, these initializations are the same for all scenes, independent from the actual object geometry/initial object state).
> > Therefore, the LGP tree consists of a total of 20 leaf nodes, 4 leaf nodes for single push, and 16 two pushes.
> > No random restarts are required for this pushing scenario.
> > The paradigm of LGP is successful if such a discrete set of decisions leads to an optimization problem that converges nicely.
> > For the mug hanging scenario, it is less clear how such discrete decisions as for the pushing intiailization could be introduced that are general and help the optimization.
> > Therefore, we addressed the local minima with random initializations of the mug within the bounding box around the hook, which we have shown works well.
> >
> > Due to your comment, we have added experiments that contain robots, which further illustrate the LGP nature of our proposed framework.
> > For example, in this experiment, the search involves which robot to use at which phase of the motion to push the object to the goal.
> > Adding robots does not require changing the problem formulation or the dynamic models.
> > The optimization problem now optimizes the joint angle movements of the robots and the motion of the pushed object simultaneously/jointly in one optimization problem, i.e. it is not the case that first the pushing is solved and then later the robot is added.
> > Therefore, the formulation can directly take the kinematic reachability limits of the robots into account when searching for a solution.
> > Another example has the goal of one robot establishing contact with the object which requires another robot to push the object towards the first robot.
> > In this case, the goal specification is only the contact establishment, not a goal location for the object. The LGP tree search together with the trajectory optimization then finds out that the other robot has to push the object towards the first robot.
> >
> > >W5. The presentation of 2a-c is too notation-heavy. Please simplify it for readability.
> >
> > We agree that it is notation heavy, but thought it to be necessary when we want to stay precise.
> > The issue is that we have explain that a discrete variable selects not only which constraints are active at which phase of the motion, but also which objects (SDFs) at which time slices are involved in those constraints. Since potentially multiple different objects at potentially multiple different times can be formulated, the notation has to reflect that somehow. Further, the discrete variable also has to obey constraints itself.
> >
> > >W6*. SDFs are clearly promising for the dynamics prediction part of the problem. This is evident from the pushing experiments. However, they introduce an inner optimization problem when establishing contact between objects (eqn 10), so it is not clear why the added complexity and cost of including an optimization problem as a constraint is worth it, compared to other representations.
> >
> > Thanks for pointing this out. We originally had an explanation of this in the main paper, but had to move it to the appendix due to space constraints.
> > We now have made an own section in the appendix for this to make it more prominent.
> > To summarize, one can rewrite that functional such that no inner optimization problem is needed by introducing an additional optimization variable.
> > This leads to a formulation without inner optimization and it converges nicely, since the SDFs provide good gradients towards their surfaces.
> >
> > When representing objects as meshes, formulating such a contact establishment constraint would typically require a convex decomposition of the objects for a collision engine and then a similar problem formulation (while the collision engine internally does some inner loop computation).
> > The SDF formulation is more direct in this case.

---

> > > ### Author Response · Authors · 2021-08-31
> > > **continue**
> > >
> > > > [...] current experiments are too simple for the paper to have any impact, but it is enjoyable to read. It would be appreciated if more challenging planning environments could be included, possibly with more objects
> > >
> > > We believe that, compared to other works that learn such models we consider, our experiments are already quite challenging and show capabilities beyond related approaches.
> > > We show generalization to multiple shapes and multi-phase manipulation planning with learned models.
> > > As mentioned before, we agree that we have not fully shown the potential of the framework, therefore, we added robots, obstacles and three interacting objects.

---

### Official Review · Reviewer_5kbt · 2021-08-08

**Originality:** Good
**Technical Quality:** Good
**Clarity Of Presentation:** Very Good
**Impact:** 4

**Recommendation:**

Weak Accept: I recommend accepting the paper, but will not argue for my recommendation if the majority of other reviewers have a different opinion.

**Summary:**

The work proposes a TAMP framework that leverages functionals of SDF-based representations of objects. Learned functionals encode both dynamics constraints (based on one-step prediction) as well as kinematic success detectors, while analytic functionals encode problem constraints such as collision avoidance and contact establishment. The motivation of such a representation is twofold: (i) SDFs are a natural vision-based object representation, and (ii) models that reason using SDFs account for full continuous object geometry, as opposed to relying on pre-specified articulated points/surfaces.

A mixed discrete-continuous optimization-based planner leverages the various functionals where the discrete optimization variable indexes “active” functionals and their relevant SDF operands, while the continuous variables correspond to the sequence of SE3 transformations on each SDF (object). Experiments are presented on two environments: (i) a static mug-hanging task, and (ii) a quasi-static pusher task.

**Issues:**

From above, limitations (2) and (3) will likely involve a lot more work, and may not be suitable for inclusion within the revision period. However, limitations (1) and (4) (or at least (1) with some discussion on (4)) would be sufficient.

**Reviewer Expertise:**

Very good: Comprehensive knowledge of the area

**Strengths And Weaknesses:**

The motivation of using SDFs as the base object representation for dynamics models is compelling and the paper presents an encouraging attempt at a model-based planning framework utilizing such a representation. In the spirit of this novelty, I recommend that the paper is accepted. However, the authors may wish to address some of the following concerns:

1. Presumably, a potential advantage of using SDFs as the object representation is that one would be able to generalize to unseen objects, with similar *local* geometric features. The experiments involved training and evaluation over the same homotopy of objects, with variation only in size. Some experiments involving different object homotopies (shapes) would be extremely interesting.

2. The notion of a “dynamic” task is somewhat underwhelming - under the the author’s own classification, this is a quasi-static task. An example involving dynamic manipulation therefore would be more insightful to showcase the capabilities of SDF-based dynamics models.

3. The manipulation examples presented in the work constitute *passive* tasks - placing a mug on a handle, or pushing an object on a stable plane. Many of the challenging examples in manipulation involve at the very least, picking up/grasping and object and then performing a sequence of task-specific transformations. In such tasks, the *stability* of grasps and subsequent manipulations comes into play. Can the SDF representation handle such tasks or would one additionally require non-trivial controllability extensions? This is closely related to the previous point on dynamic tasks.

4. Computationally, there is a natural limitation to how SDFs are evaluated and processed through dynamics models - using pre-set grids. What then is the trade-off in computation speed vs accuracy of the learned model as a function of the grid resolution? Perhaps an encoder/decoder model could be jointly trained to reduce the dimensionality of the SDF representation used within the various functionals.

**Summary Of Recommendation:**

The motivation of using SDFs as the base object representation for dynamics models is compelling and the paper presents an encouraging attempt at a model-based planning framework utilizing such a representation. In the spirit of this novelty, I recommend that the paper is accepted. However, the main drawback concerns the imitations of the experiments. Two static/quasi-static manipulation examples are presented that are passively stable. Arguably, some of the more impact-generating examples would constitute dynamic manipulation tasks, involving at the very least, picking up objects. Potential for impact is high, given the elegance of SDF representations, however more compelling experiments are needed.

---

> ### Author Response · Authors · 2021-08-31
> **Thanks a lot for your detailed review! We have added shape generalization experiments**
>
> Thanks a lot for your detailed review!
>
> >Presumably, a potential advantage of using SDFs as the object representation is that one would be able to generalize to unseen objects, with similar local geometric features. The experiments involved training and evaluation over the same homotopy of objects, with variation only in size. Some experiments involving different object homotopies (shapes) would be extremely interesting.
>
> This is a very interesting point, thanks for raising it. We have added experiments for the pushing scenario that show generalization for shapes beyond the training distribution.
> We show that the model not only maintains a high prediction performance, but the results also show that the learned model generalizes well enough in order to plan successful pushing trajectories for those out-of-distribution shapes.
> In another experiment where three objects interact with each other, a non spherical object serves as the pusher, showing additional generalization capabilities.
> Thanks again for this suggestion!
>
> >The notion of a “dynamic” task is somewhat underwhelming - under the the author’s own classification, this is a quasi-static task. An example involving dynamic manipulation therefore would be more insightful to showcase the capabilities of SDF-based dynamics models.
>
> The term "dynamic model" is commonly used for quasi-static 1st order dynamics.
> We believe that the framework could handle other dynamic models such as bouncing balls etc. as well. In this case, when the dynamics becomes 2nd order, one would have to input two SDF observations for an object at time $t-2$ and $t-1$ to the model.
>
> >The manipulation examples presented in the work constitute passive tasks - placing a mug on a handle, or pushing an object on a stable plane. Many of the challenging examples in manipulation involve at the very least, picking up/grasping and object and then performing a sequence of task-specific transformations. In such tasks, the stability of grasps and subsequent manipulations comes into play. Can the SDF representation handle such tasks or would one additionally require non-trivial controllability extensions? This is closely related to the previous point on dynamic tasks.
>
> We decided to study the mug hanging task, since it involves the challenge that the interaction between two objects (hook and mug) of different shapes have to be taken into account, while grasping usually assumes the same gripper geometry.
> Furthermore, grasping is a well studied problem, while the hanging task has not received as much attention.
>
> We believe that the framework is capable of learning whether an interaction between a gripper (represented as an SDF) and an object leads to a stable grasp.
> The fact that the success functionals model manifolds and not only a single configuration would in principle enable the system to choose grasp poses that are consistent with later manipulation steps.
> The fact that the model learns manifolds could also be seen in the mug-hanging scenario where different ways of hanging the mug on different parts of the hook are found as solutions.

---

> > ### Author Response · Authors · 2021-08-31
> > **continue**
> >
> > >Computationally, there is a natural limitation to how SDFs are evaluated and processed through dynamics models - using pre-set grids. What then is the trade-off in computation speed vs accuracy of the learned model as a function of the grid resolution? Perhaps an encoder/decoder model could be jointly trained to reduce the dimensionality of the SDF representation used within the various functionals.
> >
> > This is an important point which we want to discuss in detail.
> > The class of dynamic prediction models we propose is in some sense an encoder/decoder model.
> > The kinematic success models contain an encoder as well.
> >
> > In order to model the dynamics/kinematic success for a large variety of shapes, the model potentially has to take the \emph{whole} geometry of the objects that interact into account.
> > Furthermore, the model has to take translated and rotated objects as input as well.
> > Both is achieved by evaluating the object representations, i.e. the SDFs, on a grid.
> > This grid evaluation is then passed through an encoder (convolutional architecture).
> > In case of the kinematic success model, this encoding is used to classify success.
> > For the dynamics prediction model, based on the encoding, the model predicts the signed distance function at the next time step, which is the decoder.
> > The decoder, however, does not predict on the static grid values where the SDFs have been evaluated only, but can be queried everywhere, since it is a function and hence can interpolate.
> > Representing the dynamics this way also makes the whole approach interpretable and everything operates in this common SDF space.
> > As an alternative, one could consider encoding the SDFs and then formulating the forward dynamics as a constraint on the encoding vector. However, this would still require the same encoding step as we do.
> >
> > Computationally, for the comparisons experiments, it turned out that a pointnet architecture for encoding point-clouds is much slower than the grid evaluation and convolutional encoding with the SDFs, since GPUs have been and will continue to be optimized for such operations with very high parallelizability.
> > An other advantage of signed-distance values in such a grid compared to only occupancy indicators is that the surface can more accurately be represented even with a lower resolution, since the distance values provide more information about the object.
> >
> > In our current implementation, we query the SDFs for the learned functionals as well as for the analytic ones in the whole workspace.
> > We believe that there is plenty of room for improvement here, for example by querying the SDF not in the whole space for the collision avoidance functional, but only in the relevant part of the workspace, which would already greatly reduce the number of grid points or allows for more local accuracy.
> > Furthermore, one could think of using a fully convolutional architecture for the dynamics model, such that the encoder/decoder can operate on different grid sizes, enabling the model to be queried also only in the relevant workspace, increasing both accuracy and computational efficiency.
> > This is realizable since we predict the dynamics in SDF (or observation) space, instead of a latent space, where the dimensionality can not simply be adjusted.

---

> > > ### Author Response · Authors · 2021-09-04
> > > **Concerns addressed?**
> > >
> > > We would like to thank you again for your very valuable review.
> > > We were wondering if you had the chance of looking at the new experiment addressing the point (1) of your review regarding shape generalization (presented in section C.1, Fig. 7, Table 6, and video starting at 2:54) as well as our other responses to your review and additional experiments in the paper?
> > >
> > > Thanks!

---

### Author Response · Authors · 2021-08-31
**Response to reviewers and updated paper**

We thank the reviewers for their very helpful comments!
In summary, we added the following experiments and extensions to the paper:
* Comparisons to other object representations (occupancy measure and point-clouds) for learning dynamic models (pushing experiment)
* Comparison to planning with fully analytic models for the pushing experiment
* Comparison to occupancy measure and point-cloud object representations for mug hanging scenario
* Simultaneously learning the SDF conditioned on an image and the dynamics model
* Discussion/experimental demonstration why the SDF representation leads to better planning capabilities compared to point-cloud or occupancy measure based representations
* Generalization to out-of-distribution shapes for pushing experiments
* More complex experiments including scenes with obstacles, three interacting objects (all without retraining) as well as two collaborating robots in the scene that highlight the TAMP capabilities of the framework

With these extensive additional experiments and discussions, we hope to address the concerns of the reviewers of missing comparisons and showing that the framework as presented can handle more complex scenes/tasks.
We want to stress that the methodology and message of the paper is the same as in the original submission, the claims are just backed up with necessary comparisons and further demonstrations.
We also want to note that the problem formulation for the generalization experiments (robots, three objects, obstacles) remains the same, showing the versatility of the approach.

Changes are indicated by blue colored text in the paper.
Due to space limitations, most of these additions are presented in the appendix, however, we reference to the relevant appendix sections in the paper.

---

> ### Author Response · Authors · 2021-08-31
> **Video update**
>
> We have updated the video.
> The new results start at 2:45

---

### Meta-Review · Area_Chair_zL8B · 2021-08-17

**Recommendation:** Accept (Poster)
**Confidence:** 5

**Metareview:**

This paper proposes the use of implicit SDF representations to enable planning with task constraints using learned dynamics functions on the object SDFs. The paper validates this approach with several planning tasks in simulation including bimanual robot experiments.

The paper motivates the use of SDFs clearly for planning and shows the ability to learn accurate transition models using the SDF representation including learning the SDFs from segmented imagery. It further shows that SDFs perform better when compared to point cloud or occupancy-based binary representations.

The authors made substantial improvements to the paper based on the initial reviews and I find no lingering major issues with the work.

---

> ### Author Response · Authors · 2021-08-31
> **Thanks a lot. Many comparison experiments added**
>
> Thanks a lot for clearly pointing out where you believe the weaknesses of the paper are, which helped us focus our work on the extensions.
> We would like to mention that the goal of this paper is not solely to present another method to learn dynamic models, but to learn models that are suitable for (optimization) based planning while being able to take the geometry of the involved objects into account.
> We strongly believe that model learning and subsequent planning should be considered together, as we attempt to do in this work.
> This being said, we totally agree that the initial submission was lacking comparisons to other representations and that we could have highlighted the planning capabilities of the framework better.
> We address both by adding multiple comparisons to other approaches and more complex planning scenarios including generalization to three interacting objects, obstacles and robots in the scene.
> For those more complex planning scenarios, we want to mention that no changes in the methodology (or relearning) are required.
>
>
> >[...] it fails to compare to alternative representations in the experiments provided. Why are no experiments performed with learning directly from point clouds or meshes? To show the benefit of the SDF not only should it be shown that the representation learns better and not only that it is feasible to do learning with the SDF.
>
> We agree that the initial submission did not put enough emphasis on comparisons to other object representations (both experimentally and explanatory), in part due to severe space limitations.
> We therefore have added multiple comparisons to learning models based on point-cloud and occupancy measure object representations (the latter had been in the appendix already in the initial submission, but neither mentioned nor discussed in the main text).
>
> Results show that our proposed approach based on SDFs outperforms all other representations in prediction performance (pushing scenario) and achieved success rate (mug hanging scenario).
> We want to mention that we put more effort in tuning the hyperparameters for the point-cloud model to achieve the results than for our approach in case of the mug hanging scenario.
>
> Since our proposed formulation predicts in SDF space, the dataset contains only SDF observations. Predicting particle movements for point-clouds is less straight-forward. Therefore, in order to do these comparisons, for the point-cloud representation, we additionally need to assume to have access to ground-truth relative transformations (which requires point-cloud registration steps for the training data).
>
> Finally, we demonstrate that neither the point-cloud nor the occupancy measure representation provides useful gradients for planning, which makes them incapable of solving the complex multi-phase pushing tasks we consider.
>
>
> >Indeed learning dynamics with graphs/meshes is a popular area recently, for example "Learning Mesh-Based Simulation with Graph Networks", Tobias Pfaff, Meire Fortunato, Alvaro Sanchez-Gonzalez, Peter W. Battaglia.
>
> Thanks for pointing us to this work. While definitively interesting, they only consider learning a simulator to predict passive dynamics in a quite different setup than we do, there is no planning with the learned models.
> We have added the reference to the paper.
>
> >As such, without a direct comparison [...], the importance of this paper over existing techniques is difficult to judge. 	I would ask that the rebuttal strongly argue for why learning with perfect model-derived SDFs [...] without comparison to learning with different state representations warrants inclusion in the conference.
>
> As mentioned above, we have added multiple comparisons/experiments highlighting the advantages of the SDFs, both for learning models and later planning with them.
> With these additions, we think that we have shown the contributions of this paper over existing techniques.
>
>
> >[...] use of SDFs estimated from real sensor data [...]
>
> Based on your comment, we have investigated to learn the SDFs from image observations (with object segmentations). Both the SDFs and the dynamic model on top of them are learned simultaneously.
> The results indicate that there is only an insignificant drop in prediction performance when using learned SDFs estimated from sensor data (image).

---

> > ### Author Response · Authors · 2021-08-31
> > **continue**
> >
> >
> > >[...] without the use of any actual robot skills [...]
> >
> > We have added experiments that involve two robots in the scene for the pushing scenario.
> > The framework directly generalizes to these scenarios without relearning or having to change the methodology.
> > The planning problem is now the joint problem of planning the robot motions to achieve the desired goal, with the dynamic constraint of the interacting objects represented with the learned model based on the object SDFs.
> > This way, the planner can take the kinematic limits of the robots into account while planning for the pushing movements.
> > Furthermore, since the motions of the pushed object and the robots are planned jointly, we can address problems where the goal is that one robot tries to touch the object, but requires the other robot to push the object towards the first robot. In this case, no object goal region has to be defined, the fact that the object is being pushed towards the reachable area of the robot is fully determined by the optimization problem.
> >
> > >Specifically, what key insights can we learn from this over existing techniques beyond simply, "one can learn dynamics with SDF models"
> >
> > As mentioned, this work is not solely about "one can learn dynamics with SDF models".
> > The key insights are
> > * SDFs enable to learn a variety of models with higher performance than comparable approaches
> > * The learned models generalize to different shapes
> > * Representing models using SDF object representations provides useful gradients for trajectory optimization, enabling planning in complex scenarios for multi-step/mulit-phase manipulation
> > * The SDF representation enables to incorporate important constraints for manipulation planning such as collision avoidance and contact establishment
> > * Modeling the dynamics in SDF space enables to take the interaction of objects with different geometries into account and leads to interpretable predictions
> >
> > >[...] the prevalence of SDFs in graphics and vision research and the recent explosion of implicit representations in learning for reconstruction, rendering, and tracking (i.e. NeRF and related methods).
> >
> > This is one of the main reasons why we believe this work is very valuable for the community, since we show that SDFs not only enable to learn dynamic and kinematic success models, but also, importantly, that one can elegantly formulate a motion planning problem with them as well.
> > This complements and strengthens the results from the computer vision community.

---

### Decision · Program_Chairs · 2021-09-13

**Decision:**

Accept (Poster)

**Comment:**

This paper proposes the use of implicit SDF representations to enable planning with task constraints using learned dynamics functions on the object SDFs. The paper validates this approach with several planning tasks in simulation including bimanual robot experiments.

The paper motivates the use of SDFs clearly for planning and shows the ability to learn accurate transition models using the SDF representation including learning the SDFs from segmented imagery. It further shows that SDFs perform better when compared to point cloud or occupancy-based binary representations.

The authors made substantial improvements to the paper based on the initial reviews and I find no lingering major issues with the work.